# Androgen Receptor: A New Marker to Predict Pathological Complete Response in HER2-Positive Breast Cancer Patients Treated with Trastuzumab Plus Pertuzumab Neoadjuvant Therapy

**DOI:** 10.3390/jpm12020261

**Published:** 2022-02-11

**Authors:** Jiayi Li, Shuang Zhang, Chen Ye, Qian Liu, Yuanjia Cheng, Jingming Ye, Yinhua Liu, Xuening Duan, Ling Xin, Hong Zhang, Ling Xu

**Affiliations:** 1Breast Disease Center, Peking University First Hospital, Beijing 100034, China; lijiayi12@hotmail.com (J.L.); 06744@pkufh.com (Q.L.); chengyuanjia@163.com (Y.C.); md_ye@126.com (J.Y.); liuyinhua7520@163.com (Y.L.); xuening666@126.com (X.D.); kfs_xl@126.com (L.X.); 2Department of Pathology, Peking University First Hospital, Beijing 100034, China; candida2008@hotmail.com (S.Z.); zhanghong-path@pkufh.com (H.Z.); 3School of Public Health, Peking University Health Science Center, Haidian District, Beijing 100191, China; 1510306235@pku.edu.cn

**Keywords:** breast cancer, androgen receptor, trastuzumab, pertuzumab, neoadjuvant therapy, pathologic complete response

## Abstract

(1) Background: Neoadjuvant therapy is the main therapeutic strategy for human epidermal growth factor receptor 2 (HER2)-positive breast cancer patients, and the combination of trastuzumab and pertuzumab (HP) has become a routine treatment. How to predict and screen patients who are less likely to respond to neoadjuvant therapy is the focus of research. The androgen receptor (AR) is a biomarker that is widely expressed in all breast cancer subtypes and is probably related to treatment response and prognosis. In this study, we investigated the relationship between AR expression and treatment response in HER2-positive breast cancer patients treated with HP neoadjuvant therapy. (2) Methods: We evaluated early breast cancer patients treated with HP neoadjuvant therapy from Jan. 2019 to Oct. 2020 at Peking University First Hospital Breast Cancer Center. The inclusion criteria were as follows: early HER2-positive breast cancer patients diagnosed by core needle biopsy who underwent both HP neoadjuvant therapy and surgery. We compared the clinical and pathological features between pathological complete response (pCR) and non-pCR patients. (3) Results: We included 44 patients. A total of 90.9% of patients received neoadjuvant therapy of taxanes, carboplatin, trastuzumab and pertuzumab (TCHP), and the total pCR rate was 50%. pCR was negatively related to estrogen receptor (ER) positivity (OR 0.075 [95% confidence interval (CI) 0.008–0.678], *p* = 0.021) and positively related to high expression levels of AR (OR 33.145 [95% CI 2.803–391.900], *p* = 0.005). We drew a receiver operating characteristic (ROC) curve to assess the predictive value of AR expression for pCR, and the area under the curve was 0.737 (95% CI 0.585–0.889, *p* = 0.007). The optimal cutoff of AR for predicting pCR was 85%. (4) Conclusion: AR is a potential marker for the prediction of pCR in HER2-positive breast cancer patients treated with HP neoadjuvant therapy.

## 1. Background

Neoadjuvant therapy is the main therapeutic strategy for human epidermal growth factor receptor 2 (HER2)-positive breast cancer patients; it can downsize the lesion for surgery, provide information on treatment response and prognosis and help to adjust follow-up treatment strategies to promote survival in non-pathological complete response (non-pCR) patients [1]. At present, neoadjuvant therapy using trastuzumab and pertuzumab (HP) is commonly adopted for HER2-positive patients. Although most patients respond to this therapy, with a promising pathological complete response (pCR) rate of over 50% [2,3,4,5,6], there are some patients who do not benefit as much from this therapy. How to predict a patient’s response to neoadjuvant treatment and how to identify patients who do not respond very well to the treatment are the focus of current research.

The androgen receptor (AR) can be detected in most breast cancers, but both its expression level and prognostic effect vary. In triple-negative breast cancers (TNBCs), the luminal AR subtype expresses high levels of AR and shows decreased relapse-free survival [7,8,9,10]. In hormone receptor (HR)-positive breast cancers, AR expression is associated with better outcomes [10,11,12,13]. However, in HER2-positive breast cancers, the relationship between AR and survival is unclear. In HER2-enriched metastatic breast cancer, AR-positivity with a cut-off value of 10% was associated with both longer progression-free survival (PFS) and an increased overall survival (OS) rate and could also predict the efficacy of first-line trastuzumab treatment [14]. AR positivity was also associated with significantly better OS in AR+ non-luminal HER2 positive breast cancer treated with neoadjuvant chemotherapy [15]. However, a higher AR mRNA level was associated with worse disease outcomes in HER2-positive tumors [11]. Another analysis showed that although AR positivity was associated with favorable clinical outcome in the entire early-stage breast cancer, AR positivity was not significantly associated with disease-free survival (DFS) and was significantly associated with a worse OS rate in the HER2 positive breast cancer subgroup [16]. Molecular apocrine tumors (estrogen receptor [ER] negative and AR positive) with HER2 positive status also showed a poor outcome [12]. There are also studies that show no relationship between AR and outcome. However, a study of 4147 patients, which followed them for a median 16.5 years, showed that AR was not associated with survival among ER+/HER2+ tumors or ER−/HER2+ tumors [10]. Studies show that AR can probably predict neoadjuvant treatment effects. AR expression seemed to be related to a lower pCR rate and worse outcome in TNBC [3,17,18,19,20], but there was also controversy related to these results [21]. In HER2 positive breast cancer, AR+ positivity was associated with pCR, but in this study, only 57.3% of patients received trastuzumab [15]. Currently, no study has focused on the relationship between AR expression and the response to neoadjuvant therapy including HP in HER2-positive breast cancer patients.

There is no acknowledged cutoff for the AR expression level. Most researchers suggest adopting 1% or 10% for both ER and the progesterone receptor (PR), but a cutoff of 35% is also advised because of its prognostic value for recurrence-free survival (RFS) after surgery [22]. In our study, we explored the potential cutoff for the purpose of predicting the pCR rate of neoadjuvant therapy in HER2-positive breast cancer patients.

## 2. Materials and Methods

### 2.1. Patients

We reviewed early breast cancer patients treated with HP neoadjuvant therapy from January 2019 to October 2020 at Peking University First Hospital Breast Cancer Center. The inclusion criteria were as follows: early HER2-positive breast cancer patients diagnosed by core needle biopsy who underwent both HP neoadjuvant therapy and surgery. The exclusion criteria were as follows: (1) metastatic breast cancer; (2) failure to complete the scheduled treatment; and (3) missing AR expression and unable to be tested (Figure 1). This study is reported according to the reporting recommendations for tumor MARKer prognostic studies (REMARK) criteria [23].

### 2.2. Specimen Preparation and Hematoxylin and Eosin (HE) Staining

The protocol of tissue handling was standardized according to the recommendations of the 2007 American Society of Clinical Oncology (ASCO)/College of American Pathologists (CAP) guidelines [24]. The breast tissue samples were fixed in 10% neutral buffered formalin, dehydrated through a serial alcohol gradient, and embedded in paraffin wax blocks. The sections were then stained with HE. Two experienced pathologists reviewed and distinguished the histological subtype and grading based on the World Health Organization (WHO) classification of breast cancer and Nottingham grading [25].

### 2.3. Immunohistochemistry

Biomarkers of ER, PR, HER2 and AR were detected in all the samples using the immunohistochemical stainer Ventana Benchmark XT for HER2 and Dako Autostainer Link 48 for other markers. The antibodies were ER (1D5, Dako, at 1:50 dilution), PR (636, Dako, at 1:200 dilution), HER2 (4B5, Ventana), Ki67 (MIB1, Dako, at 1:100 dilution) or AR (EP120, GBI). According to the 2010 ASCO/CAP guidelines [26], ER or PR was positive if ≥1% of tumor cells were immunoreactive and negative if <1% of tumor cells were immunoreactive. HER2 expression was determined according to the 2013 ASCO/CAP guideline [27]. A score of 3+ was regarded positive, and a score of 0–1 was negative. A score of 2+ was interpreted as equivocal and was tested with fluorescence in situ hybridization (FISH). According to the 2018 ASCO/CAP guidelines [28], a HER2/CEP17 ratio ≥ 2.0 with a HER2 signals/cell ratio ≥ 4.0 or a HER2/CEP17 ratio < 2.0, with a HER2 signals/cell ≥ 6.0, were positive. The molecular subtypes were differentiated according to the 2011 St. Gallen International Expert Consensus: Luminal B HER2 positive (ER+, PR+ or −, HER2+ and Ki-67 ≥ 15%) and HER2-overexpressed (ER-, PR- and HER2+) [29]. Then, the expression of AR was described by the proportion and intensity of positive staining of tumor cell nuclei. Ki-67 was evaluated by the percentage of tumor cell nuclei with positive immunostaining. Based on the Nottingham combined histologic grade [25], histological scores and grades were made according to adeno-tube formation, nuclei size and shape, chromosome heterogeneity and nuclear division. Anatomical staging was performed according to the 8th edition of the American Joint Commission on Cancer (AJCC) breast cancer staging system [30].

### 2.4. Neoadjuvant Therapy and Evaluation of Efficacy

In our study, the scheduled neoadjuvant therapy was six cycles of taxanes, carboplatin, trastuzumab and pertuzumab (TCHP) as follows: (1) T: docetaxel 75 mg/m^2^ administered by intravenous infusion every 3 weeks or albumin-bound paclitaxel 200–260 mg/m^2^ administered by intravenous infusion every 3 weeks; (2) C: carboplatin, total dose (mg) = Area Under Curve(AUC) × (glomerular filtration rate + 25) (AUC is 4–5), administered by intravenous infusion every 3 weeks; (3) H: trastuzumab 8 mg/kg administered by intravenous infusion for the first time and 6 mg/kg administered by intravenous infusion afterward every 3 weeks and (4) P: pertuzumab 840 mg administered by intravenous infusion for the first time and 420 mg administered by intravenous infusion afterward every 3 weeks.

The clinical response was evaluated according to Response Evaluation Criteria in Solid Tumors (RECIST) 1.1 [31]. Patients underwent surgery after scheduled neoadjuvant therapy. The Miller–Payne system was used to assess the effect of neoadjuvant treatment [32], and the pathological response was evaluated according to Collaborative Trials in Neoadjuvant Breast Cancer (CTNeoBC). pCR was defined as the absence of invasive cancer in the breast and axillary nodes, irrespective of ductal carcinoma in situ [33].

### 2.5. Statistical Analysis

The clinicopathological characteristics of the groups were compared by using the independent-sample *t* test for continuous variables, Pearson χ^2^ test or Fisher’s exact probability test for categorical variables and the Mann–Whitney U test for grade variables. Relevant factors were analyzed by logistic regression. The diagnostic test was analyzed by receiver operating characteristic (ROC) curve analysis. *p* < 0.05 indicated that the difference was statistically significant. All tests were two-sided tests. All analyses were performed with IBM SPSS 26.

## 3. Results

### 3.1. Patients

A total of 44 patients were enrolled in our study. All the patients were women. The median age was 46.5 years, and the Eastern Cooperative Oncology Group (ECOG) performance status was 0–1. Of the patients, 23 (52.3%) had the HER2-overexpressed subtype and 21 (47.7%) had the luminal B HER2-positive subtype. Of the 21 Luminal B HER2 positive patients, 6 patients were ER+ PR- and 15 were ER+ PR+. One patient was in stage T1, 36 patients were in stage T2 and 7 patients were in stage T3. A total of 22 patients (50%) were in stage N1. Five patients had Ki-67 <30% and 38 had Ki-67 ≥ 30%. Six patients did not express AR. Thirteen patients had an AR expression level of 50–89% and eighteen had an AR expression level ≥90%. Forty patients received TCHP neoadjuvant therapy and four patients received other therapies (Table 1).

### 3.2. Factors Influencing Neoadjuvant Therapy Response

We performed multiple logistic regression including factors such as age, Body Mass Index (BMI), tumor stage and molecular biomarkers. AR expression over 90% was the only statistically significant factor for pCR (OR 729.322, 95% Confidence Interval, CI 1.778–299,130.165, *p* = 0.032). We then used the backward LR method to select the factors. It showed that AR expression over 90% was positively correlated with pCR (OR 33.145, 95% CI 2.803–391.900, *p* = 0.005) and that ER positivity was negatively correlated with pCR (OR 0.075, 95% CI 0.008–0.678, *p* = 0.021). Age, BMI, tumor stage, histological grade, PR, HER2 and Ki-67 were not significantly related to pathological response (Table 2 and Table 3).

### 3.3. ROC Curve Analysis of AR for the Prediction of pCR

ROC curve analysis of AR expression for the prediction of pCR was performed (Figure 2). The AUC was 0.737 (95% CI 0.585–0.889, *p* = 0.007). This result indicated that as the AR expression level increased, the patient was more likely to have pCR. To maximize sensitivity and specificity, 85% was the best cutoff of AR expression level to predict pCR. The sensitivity was 0.636, and the specificity was 0.818.

## 4. Discussion

According to the St. Gallen Conference of 2013, breast cancer can be classified into subtypes according to molecular expression [29]. The two subtypes of HER2 positivity are the HER2-overexpressed and luminal B HER2-positive subtypes, accounting for 15–20% of all breast cancer cases. HER2 modulates cell differentiation, apoptosis and other activities through pathways such as PI3K/AKT and RAS/RAB/MEK/MAPK [34]. Anti-HER2 targeted therapy is growing quickly and is widely used in neoadjuvant treatments, adjuvant treatments and salvage treatments. Clinical trials have shown the promising effect of HP in neoadjuvant treatments, but pertuzumab was inaccessible in China until 2019, when it was included in healthcare insurance. To date, there have been few studies on Chinese breast cancer patients treated with HP.

AR widely exists in all breast cancer subtypes. Studies of AR have mostly focused on the TNBC subtype, but few have focused on the HER2-positive subtypes. In HER2-positive breast cancer cells, AR promotes the expression of the HER3 gene, and the HER2/HER3 heterodimer activates the PI3K/AKT pathway and the MYC gene and promotes cell proliferation. HER2 can also activate AR transcription and ERK, which in turn modulates HER2 and AR [35,36]. An additional subtype of molecular apocrine breast cancer was identified in recent microarray studies. This subtype features ER negativity, AR positivity and apocrine differentiation, and is also associated with HER2 amplification [37,38,39]. AR is highly expressed in HER2 enriched (HER2E) breast cancer, and HER2E breast cancers are hormonally driven, either by ER in ER+ HER2E tumors or by AR in ER- HER2E tumors [40]. AR pathway activity and AR expression are also found to be positively correlated in HER2-positive breast cancer and inversely correlated in HER2-negative breast cancer [41]. These studies show a crosstalk between AR and HER2.

In our study, the median age of the enrolled patients was 46.5 years, which was consistent with the characteristics of early age of onset in China. Approximately 70–90% of luminal breast cancers and 60–70% of HER2-positive breast cancers express AR [42,43,44,45,46,47,48,49,50]. In our study, the proportion of breast cancers with AR expression was 86%, which matched the literature. All patients received full-course neoadjuvant therapy, including HP, and the pCR rate was 50%. However, in the literature, the pCR rate of neoadjuvant therapy, including trastuzumab and pertuzumab, was 54–67% [2,3,4,5,6], which was higher. We inferred that this was due to the difference in age and regions and that the small population could also cause bias. A larger study is warranted to determine the true pCR rate of HER2-targeted neoadjuvant therapy in China. In the previous studies, pCR was related to histological grade, ER, PR, Ki-67 and lymph node status [51,52,53]. However, we found that only high AR expression and ER negativity were significantly related to pCR. In our study, the population was rather small, and only five patients had Ki-67 expression < 30%. This could probably explain why Ki-67 was not significantly associated with pCR in our analysis.

AR expression has been suggested to predict the response to neoadjuvant therapy in the literature. AR-negative TNBC patients had higher pCR rates than AR-positive TNBC patients [3,18,19,20]. In these studies, researchers often adopted 1% or 10% as the cutoff of AR. In a study of 82 HER2-positive breast cancer patients treated with neoadjuvant therapy [15], AR scores ≥ 4 were regarded as AR positive. Although only 57% of patients received HER2-targeted treatment, the pCR rate was as high as 49%. AR-positive and non-luminal types were both associated with pCR. For survival analysis, AR positivity was associated with better OS and DFS in several subgroups. Our result agreed with the literature. There are other studies focusing on the cutoff of AR. AR > 78% has been shown to predict the survival of ERα-positive patients [54], and AR > 35% has been shown to predict RFS in patients with surgically resected breast cancer [22]. In our study, we found that a cutoff value of 85% could predict the response to neoadjuvant therapy including HP. However, the raw data of AR expression level in our study were all integers of 10%, which meant that the cutoff of 85% was the dividing line between AR ≥ 90% and AR ≤ 80%. Therefore, we suggest a cutoff value of 90% as a more practical boundary line. The higher the AR expression level, the more likely it is that the patient has pCR. This cutoff could probably be used to predict and select non-pCR patients and help to adjust treatment strategies to achieve better outcomes.

The highlight of our study is that this is the earliest study on the predictive value of AR in neoadjuvant treatment with HP. Only a short amount of time has passed since pertuzumab was approved for use in China, and there are few studies on the effect of HP in the Chinese population. Our study is relatively early and significant in China at the moment. However, there are still limitations. First, the population was small and from a single center. A larger, multicenter study is needed to show the effect of HER2-targeted neoadjuvant treatment in Chinese people and the predictive value of AR. Second, pertuzumab has only been accessible in China since 2019 and some patients are still receiving adjuvant HER2-targeted therapy. The follow-up time was too short to assess long-term survival. We will continue to follow up with the patients for further study on survival.

## 5. Conclusions

Neoadjuvant therapy including HP is a major strategy for treating HER2-positive breast cancer patients. AR expression has the potential to predict the effects of treatment, but a larger population and longer follow-up are needed to show its value. AR is also a probable drug target in breast cancer. We look forward to the promising performance of AR in the treatment of breast cancer.

## Figures and Tables

**Figure 1 jpm-12-00261-f001:**
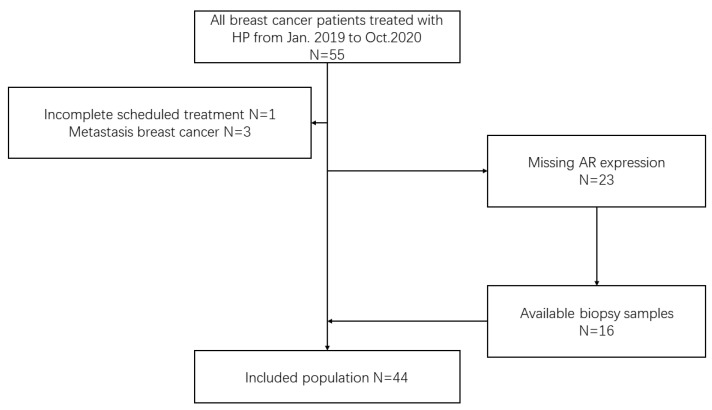
Flow diagram of the patient population. HP neoadjuvant therapy containing trastuzumab and pertuzumab, AR androgen receptor.

**Figure 2 jpm-12-00261-f002:**
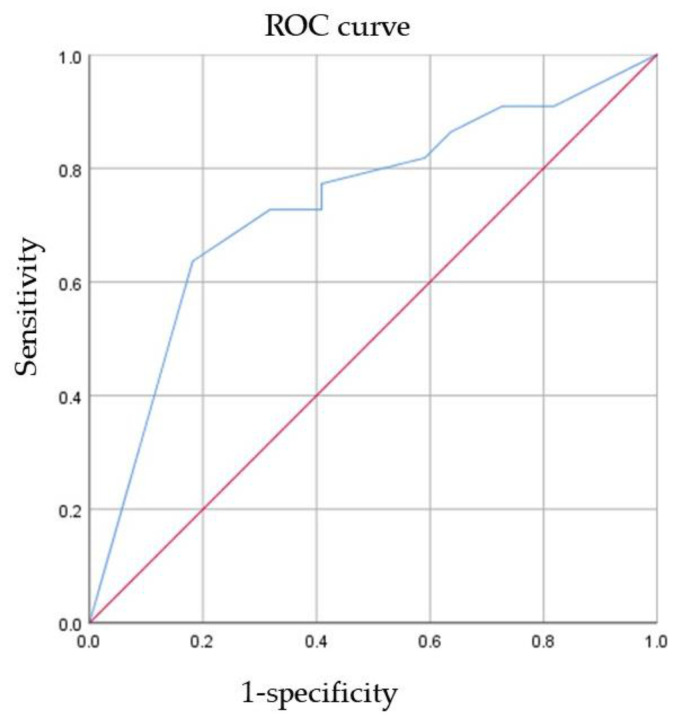
The ability of AR to predict pCR. *ROC* receiver operating characteristic, *AR* androgen receptor, *pCR* pathological complete response.

**Table 1 jpm-12-00261-t001:** Baseline characteristics of the patients. BMI Body Mass Index, ECOG the Eastern Cooperative Oncology Group, ER estrogen receptor, PR progesterone receptor, HER2 human epidermal growth factor receptor 2, FISH fluorescence in situ hybridization, *AR* androgen receptor, pCR pathological complete response, TCHP taxanes, carboplatin, trastuzumab and pertuzumab, NPHP N: vinorelbine 25–30 mg/m^2^ administered by intravenous infusion on day 1 and day 8 every 3 weeks; P: cisplatin 75 mg/m^2^ administered by intravenous infusion, completed in 2–3 days, every 3 weeks; H: trastuzumab; P: pertuzumab.

Characteristic	n = 44
Sex	
Female	44 (100%)
Male	0 (0%)
Age, years	47.95 (24–68)
BMI	23.59 (16.77–36.44)
Menopausal status	
Premenopausal	27 (61.4%)
Postmenopausal	16 (36.4%)
Unknown	1
Molecular subtype	
HER2 overexpressed	23 (52.3%)
Luminal B HER2 positive	21 (47.7%)
ECOG performance status at enrollment	
0	43 (97.7%)
1	1 (2.3%)
T stage	
T1	1 (2.3%)
T2	36 (81.8%)
T3	7 (15.9%)
N stage	
N0	22 (50%)
N+	22 (50%)
Histological grade	
2	19 (43.2%)
3	24 (54.5%)
Unknown	1
Histological score	
6	10 (22.7%)
7	8 (18.2%)
8	15 (34.1%)
9	9 (20.5%)
Unknown	2
ER	
<1%	23 (52.3%)
≥1%	21 (47.7%)
PR	
<1%	29 (65.9%)
≥1%	15 (34.1%)
ER/PR expression	
ER-PR-	23 (52.3%)
ER + PR-	6 (13.6%)
ER+ PR+	15 (34.1%)
HER2	
3+	37 (84.1%)
2+ and FISH+	7 (15.9%)
Ki-67	
<30%	5 (11.4%)
≥30%	38 (86.4%)
Unknown	1
AR	
0	6 (13.6%)
≥1%	38 (86.4%)
<50	7 (18.4%)
50–90	13 (34.2%)
≥90	18 (47.49%)
Neoadjuvant therapy	
TCHP	40 (90.9%)
NPHP	4 (9.1%)
Pathological response	
pCR	22 (50.0%)
Non-pCR	22 (50.0%)

**Table 2 jpm-12-00261-t002:** Factors influencing neoadjuvant therapy response. BMI Body Mass Index, ER estrogen receptor, PR progesterone receptor, HER2 human epidermal growth factor receptor 2, FISH fluorescence in situ hybridization, AR androgen receptor, pCR pathological complete response, TCHP taxanes, carboplatin, trastuzumab and pertuzumab, NPHP vinorelbine, cisplatin, trastuzumab and pertuzumab, OR odds ratio, CI confidence interval.

	B	Standard Error	Wald	Significance	OR (95% CI)
Age					
≤45					
>45	0.202	1.116	0.033	0.856	1.224 (0.137–10.911)
BMI					
<18.5			1.035	0.793	
18.5–24	−18.216	26,106.954	0.000	0.999	0.000
24–28	−19.632	26,106.954	0.000	0.999	0.000
>28	−39.525	30,878.473	0.000	0.999	0.000
T stage					
T1–2					
T3	3.354	2.040	2.704	0.100	28.614 (0.525–1558.715)
N stage					
N0					
N+	−0.997	1.217	0.671	0.413	0.369 (0.034–4.007)
Histological grade					
G2					
G3	−0.639	1.269	0.254	0.614	0.528 (0.044–6.344)
ER					
ER-					
ER+	−3.939	2.798	1.982	0.159	0.019 (0–4.689)
PR					
PR-					
PR+	−0.493	2.218	0.049	0.824	0.611 (0.008–47.198)
HER2					
3+					
2+ and FISH+	1.738	1.965	0.783	0.376	5.686 (0.121–267.375)
Ki67					
≤30					
>30	4.730	2.542	3.462	0.063	113.247 (0.777–16,504.22)
AR					
0–49			4.995	0.082	
50–89	1.790	1.610	1.236	0.266	5.99 (0.255–140.57)
≥90	6.592	3.070	4.612	0.032	729.322 (1.778–299,130.165)
Neoadjuvant therapy					
TCHP					
non-TCHP	−3.675	2.490	2.178	0.140	0.025 (0–3.338)
Constant	16.939	26,106.954	0.000	0.999	22,733,481.85

**Table 3 jpm-12-00261-t003:** Factors influencing neoadjuvant therapy response after selection. ER estrogen receptor, AR androgen receptor, OR odds ratio, CI confidence interval.

	B	Standard Error	Wald	Significance	OR (95% CI)
ER					
ER-					
ER+	−2.597	1.126	5.315	0.021	0.075 (0.008–0.678)
AR					
0–49			8.809	0.012	
50–89	0.034	0.972	0.001	0.972	1.035 (0.154–6.95)
≥90	3.501	1.260	7.717	0.005	33.145 (2.803–391.9)
Constant	0.039	0.714	0.003	0.956	1.04

## Data Availability

The datasets generated and analyzed during the current study are not publicly available due to patients’ individual privacy but are available from the corresponding author on reasonable request.

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
