# Peer review of "Androgen Receptor: A New Marker to Predict Pathological Complete Response in HER2-Positive Breast Cancer Patients Treated with Trastuzumab Plus Pertuzumab Neoadjuvant Therapy"

_jpm, 2022, doi:10.3390/jpm12020261_

Round 1
Reviewer 1 Report
In this manuscript the authors want to establish AR as a predictive marker for complete pCR in HER2 positive breast cancer treated with trastuzumab and pertuzumab neoadjuvant therapy. Although the authors have assessed 44 clinical samples. The manuscript lack many experimental evidences and does not establish AR as a predictive marker.
Author Response
Thank you for your suggestions on our manuscript. We learnt a lot from your suggestions. We consulted the literature and summarized the experimental evidences of the crosstalk of HER2 and AR. Genomic studies showed that AR pathway could probably drive the HER2 enriched breast cancer[1,2] (Line 223-230). We also added details on the predictive value of AR in the literature (Line 252-254). We found that AR expression over 90% was positively associated with pCR in HER2 positive breast cancer, but ER expression was negatively associated with pCR, and that other characteristic such as histological grade and Ki-67 were not associated. We also found that the cutoff of 85% was the best to predict pCR. Hence, we suggest AR as a predictive marker (Line 231-245). Indeed, more experimental evidences would better show the predictive value of AR. Because of the limitation of time, we could not complete translational experiments now. We are planning more in-depth translational research on AR in HER2 positive breast cancer. We appreciate your suggestion on our direction of future research.
Reference
- Daemen, A.; Manning, G. HER2 is not a cancer subtype but rather a pan-cancer event and is highly enriched in AR-driven breast tumors. Breast cancer research : BCR 2018, 20, 8, doi:10.1186/s13058-018-0933-y.
- Liu, D. AR pathway activity correlates with AR expression in a HER2-dependent manner and serves as a better prognostic factor in breast cancer. Cell Oncol (Dordr) 2020, 43, 321-333, doi:10.1007/s13402-019-00492-6.
Reviewer 2 Report
In the introduction section:
- This sentence "However, in HER2-positive breast cancers, the relationship between AR and survival is unclear" has to be explain. The authors should better explain the controversy of the previous works they cite, so that the reader can better understand why the relationship between AR and survival is not clear.
In the materials and methods section:
- The immunohistochemistry section needs to be more detailed to allow the reader a better understanding of the methodology used.
- Also details concerning the antibodies used (clones, suppliers etc) should be mentioned.
- The legends of all the figures should be more clear. Authors should describe what the figure represents and explain the meaning of each acronym used in the figure.
- Table 1:
The formatting of the results needs to be improved to better differentiate the characteristics. e.g. put sex, age etc in bold. - The authors should analyse the results taking into account the kind of treatment received by the patients in order to determine if the predictive value of the AR is better for a type of adjuvant therapy or not (e.g. docitaxel versus paclitaxel) because even if these two molecules are analogous they do not seem to have the same efficacy.
Author Response
In the introduction section:
- This sentence "However, in HER2-positive breast cancers, the relationship between AR and survival is unclear" has to be explain. The authors should better explain the controversy of the previous works they cite, so that the reader can better understand why the relationship between AR and survival is not clear.
Response: Thank you for your suggestions on our manuscript. Your suggestions greatly helped us.
- We added details of the controversy of the previous works in the background section (Line 57-71,73-75). We hope the readers would better understand the questions on AR.
In the materials and methods section:
- The immunohistochemistry section needs to be more detailed to allow the reader a better understanding of the methodology used.
Response: 1. We added more details on the immunohistochemistry methods we adopted (Line 96-104.)
- Also details concerning the antibodies used (clones, suppliers etc) should be mentioned.
Response: 2. We also added more details on the antibodies we used (Line 106-110).
- The legends of all the figures should be more clear. Authors should describe what the figure represents and explain the meaning of each acronym used in the figure.
Response: 3. We spelled out the abbreviations in the figure and table legend (Line 94-95, 204-206, 168-172, 187-191, 193-194) and modified the legend of Figure 2 (Line 204-206). We also explained the result more clearly in Line 179-183.
- Table 1:
The formatting of the results needs to be improved to better differentiate the characteristics. e.g. put sex, age etc in bold.
Response: 4. We made characteristic row bold and made characteristic clearer in Table 1 and we also made bold characteristic rows and added reference rows in Table 2 and Table 3.
- The authors should analyse the results taking into account the kind of treatment received by the patients in order to determine if the predictive value of the AR is better for a type of adjuvant therapy or not (e.g. docitaxel versus paclitaxel) because even if these two molecules are analogous they do not seem to have the same efficacy.
Response: 5. Thank you very much for this suggestion of analyzing the therapy. We agree that different drugs would possibly affect the response. However, in our research, the population was 44, which was too small to show the difference of the drugs. Further, the drugs were not randomly decided. Mostly we adopted docetaxel, but albumin-bound paclitaxel was adopted in some allergic and diabetes patients. We would collect more patients in the future research and make more in-depth studies on neoadjuvant therapy. Thank you again for your suggestion.
Reviewer 3 Report
In your article “Androgen receptor: A new marker to predict pathological complete response in HER2-positive breast cancer patients treated with trastuzumab plus pertuzumab neoadjuvant therapy” your preliminary data analysis shows that AR is a potential marker for the prediction of pCR in HER2-positive breast cancer patients treated with HP neoadjuvant therapy in your patient population. Your study is important and your future analysis of more patient data should confirm the predictive value of AR.
Please spell out all abbreviations such as human epidermal growth factor receptor 2 (HER2) in the first sentence in background.
Author Response
Thank you for your suggestions on our manuscript. Your comments greatly encouraged us and we would make more in-depth study on AR. We checked our manuscript and spelled out the abbreviations in the abstract and main body (Line 17, 30, 31, 42, 80, 91, 136, 176).
Round 2
Reviewer 1 Report
Thank you for clarifying and working towards improvement of manuscript. New version is adequate to endorse.
Reviewer 2 Report
The improvements made to the document seem sufficient to me.